# Has food security in the EU countries worsened during the COVID-19 pandemic? Analysis of physical and economic access to food

Karolina Pawlak [1], Agata Malak-Rawlikowska [2]*, Mariusz Hamulczuk[3], Marta Skrzypczyk[3]

1 Department of Economics and Economic Policy in Agribusiness, Faculty of Economics, Poznan University of Life Sciences, Poznan, Poland, 2 Department of Economics and Organisation of Enterprises, Institute of Economics and Finance, Warsaw University of Life Sciences, Warsaw, Poland, 3 Department of International Economics and Agribusiness, Institute of Economics and Finance, Warsaw University of Life Sciences, Warsaw, Poland

☯ These authors contributed equally to this work.
* agata_malak-rawlikowska@sggw.edu.pl

**Data Availability Statement:** The data underlying the results presented in the study are available from public databases EUROSTAT, FAO. All data

## Abstract

The aim of the paper is to provide an ex-post assessment of the impact of the COVID-19 pandemic on food insecurity in the EU-27 countries expressed by physical and economic food access. We analysed trade and price effects, together with food insecurity and malnutrition indicators. Actual levels of the indicators were compared with their pre-pandemic magnitudes and/or with counterfactual levels derived from predictive models. We also aimed to compare the objective statistics with the subjective consumers' perception of their households' food security. Our research indicates that the EU food trade was more resilient to COVID-19 impacts than the trade in non-food products, while food trade decreases were of a temporary nature. This did not affect the trade balance significantly; however, the import reduction threatened the physical food access in most EU countries. Regarding economic food access, the results indicate that the increase in food prices was offset by the increase in disposable income. It may suggest that the COVID-19 pandemic did not significantly affect the deterioration of economic access to food in the EU countries. However, the prevalence of severe food insecurity in the total population or the proportion of households reporting inability to afford a meal with meat, chicken, fish, or a vegetarian equivalent increased in 2020–2021 compared to 2019. This means that the comparative analysis of the real data on prices and households' income, as well as consumer financial situation and food consumption affordability, does not offer a clear answer concerning the impact of the COVID-19 pandemic on the food security of EU households.

are fully available without restriction under the following links: https://ec.europa.eu/eurostat/data/database. https://www.eurofound.europa.eu/surveys/european-quality-of-life-surveys/european-quality-of-life-survey-2016/eqls-2016-methodology https://www.fao.org/faostat/en/#data/FS.

**Funding:** This research has received funding from the National Science Centre within the OPUS research project no. 2021/41/B/HS4/03161 entitled "The implications of the COVID-19 crisis for the spatial integration of agri-food markets and the functioning of food supply chains in the world, with a particular focus on Poland." The funders had no role in study design, data collection and analysis, decision to publish, or preparation of the manuscript.

**Competing interests:** The authors have declared that no competing interests exist.

## Introduction

Agri-food products are among the essential goods satisfying basic human needs. Due to climate and soil restrictions, agricultural production is not evenly distributed in spatial terms (geographically), hence there is a need for exchange between self-sufficient regions and regions with food shortages. Along with technological progress, globalisation, international integration processes, and trade liberalisation processes, linkages between economies have strengthened. This has contributed to unprecedented economic growth, improved global welfare, and increased food security. However, this equilibrium is rather sensitive to emerging political, economic, or climate-environmental shocks. The latter plays an increasingly important role in leading to disturbances in the processes of spatial integration of agri-food markets and affects the functioning of local and global food supply chains, thus influencing food security on a local, regional, and global scale. One of the biggest shocks of health nature having political, socioeconomic and environmental implications we have experienced globally in the past few decades is related to the outbreak of the COVID-19 pandemic. Observations show that the COVID-19 pandemic has disrupted global supply chains in almost all sectors of economies and countries [1–11]. Restrictions on the flows of goods, services, and production inputs introduced as a result of a pandemic, leading to a weakening of the spatial integration of agri-food markets, can have serious repercussions for food security in many regions worldwide [12, 13]. According to the FAO estimates [14], between 702 and 828 million people were affected by hunger globally in 2021. The number grew by about 150 million after the outbreak of the COVID-19 pandemic: 103 million more people between 2019 and 2020 and 46 million more in 2021. In 2021, hunger affected 278 million people in Africa, 425 million in Asia, and 56.5 million in Latin America and the Caribbean. While most of the world's undernourished people live in Asia, Africa is the region where the prevalence of undernourishment is the highest [15].

However, the COVID-19 pandemic also had an impact on the economies and food markets of highly developed and food self-sufficient countries, including the EU countries. Unfortunately, this type of research has rarely been carried out for the EU, probably because the problem of food insecurity has become less and less significant there over the past decades. This is indeed true if we analyse the country's aggregated levels. However, at the household level, the situation is not clear-cut, while the level of food security varies depending on the size of the household as well as other factors [16]. According to the FAO [15] statistics, in 2021 about 14.4 million people in Europe (1.9% of the population) suffered from severe food insecurity, which doubled from 2019. The number of people who experienced moderate and severe food insecurity in Europe in 2021 amounted to 58.3 million (7.8% of the population), and increased from 2019 by 6.7 million people [15]. Those deteriorating indicators along with logistics constraints leading to the disruption in both local and international food supply chains, fuelled consumer fears concerning food availability. Thus, our task is to verify whether the COVID-19 pandemic has contributed to the deterioration of the level of food security and whether the statistics analysed are consistent with the subjective consumers' perception of their households' food security. Therefore, the scientific aim of the paper is to provide an ex-post assessment of the impact of the COVID-19 pandemic on the physical and economic access to food in the EU-27 countries, resulting in the overall state of food security.

The paper attempts to provide answers to the following questions:

1. What was the effect of the COVID-19 pandemic on the physical food access in the EU countries? This is done by examining food trade dynamics.

2. What was the effect of the COVID-19 pandemic on the economic food access in the EU countries? This is done by examining monthly and annual changes in food prices, the share

of food in inflation, citizens' financial situation, and the share of food expenditures in income. Conclusions from those analyses were supplemented by investigating changes in GDP per capita and average wage per employee.

To measure developments in physical and economic food access, we tracked as well changes in the (in)ability to afford a meal with meat, chicken and fish (or vegetarian equivalent) every second day.

According to the best of our knowledge, such analyses have been carried out neither for the European Union as a whole nor for all its member states. An additional contribution of the study to the state of the current knowledge lies in the measurement methodology used, which is based on differences between the level of variables in the years covered by the pandemic (2020–2021), but prior to the Russian-Ukrainian war (which could have made the COVID-related impact less clear), with forecasts of these variables obtained from forecasting models estimated based on data before the outbreak of the pandemic (until 2019 or until February 2020). This approach facilitates estimation of the impact of COVID-19 itself by taking into account pre-existing trends or seasonal variations. A comparison of real data with counterfactuals obtained from the forecasting models can be an alternative to simulation-based or econometric model-based approaches. Finally, the novelty of our study results from comparing real data with the consumers' perception of their food security. An objective assessment of the state of food security based on statistical data may differ from the subjective feelings of consumers. It can also be added that this study is one of the first attempts to conduct an ex-post analysis of the COVID-19 impact on food security. Most up-to-date studies were of a descriptive nature or included simulation analyses based on fragmented ex-post data.

The results from our analysis enrich the state of our knowledge concerning mechanisms of functioning of markets in the face of an external shock having the character of an epidemic threat. It is extremely important because the impact of these types of external factors is a significant determinant of the functioning of food markets and supply chains and may distort the sphere of food security in the long term.

## External shocks and food insecurity–a short literature review

Political and military conflicts, natural disasters, and epidemics are types of external shocks that can significantly disrupt the functioning of supply chains in agri-food markets and affect food security. The latter shocks include e.g. the BSE (Bovine Spongiform Encephalopathy) outbreak in Western Europe in the 1990s [1, 17], the avian flu outbreak in 2003–2006 [18], the swine flu outbreak in 2009–2010, the African Swine Fever incursion that has spread in Europe since 2007, and a current COVID-19 pandemic started in 2020 [19]. All those market distortions cause serious repercussions for food security on a local, national, regional, or global scale not only by limiting the physical availability of food but also by reducing the economic access to food (by increased prices and food expenditures), deteriorating diet quality, and interrupting supply stability.

Several past studies have examined the economic and food security effects of the abovementioned epidemics on national economies or their sectors. Jin et al. [20] and Gollamudi [17] proved significant changes in beef, pork and poultry meat prices after the BSE cases in the US. Moreover, the ban on US beef in several major importing countries affected the cattle husbandry and beef industry in the US. Bloom et al. [18] projected slowed economic growth in Asia and a significant reduction in trade as consequences of avian flu in Asia. Obayelu [21] showed that a severe impact was revealed not only in the poultry industry, but also referred to food security and the livelihoods of both rural and urban communities. The economic modelling approach was also used to examine the consequences of an African swine fever outbreak

for China and the global market [22]. The models projected global pork prices to increase by 17–85% and the unmet demand to drive price increases in the case of other meats. As far as the effects on food and nutrition security are concerned, it was proved that the projected changes in food prices would increase food expenditure, which may lead to a declining average per capita calorie availability [22]. This might be a problem mainly in middle-income countries, where calorie availability has already been low.

The COVID-19 pandemic influenced the whole national economy and food security regardless of the level of economic development. IMF [19] shows that cumulative per capita income losses over 2020–22, compared to pre-pandemic projections, are equivalent to 20% of 2019 GDP per capita in emerging markets and developing economies (excluding China), while in advanced economies, the losses are expected to be relatively smaller, at 11%. Statista Survey [23] shows that food shortages are one of the most common concerns expressed by consumers in relation to the COVID-19 pandemic, right after health, the decline in household income as a result of job loss, and the country's economic stability. For instance, food shortages were a cause for concern for 16% of the population in Germany and 36% of the population in the US. At the same time, anxiety about the economic situation and purchasing power that shapes economic access to food was expressed by 37% of respondents in Germany, or 47% and 48% of respondents in the US and China, respectively. COVID-19 has also had immediate consequences for household incomes and raised households' concerns about the capacity of the food system to ensure food availability in Canada [2].

Declining incomes and introducing bans or restrictions on exports of sensitive products, including food, had severe consequences for countries struggling with food insecurity. Laborde et al. [24] predicted an increase in extreme global poverty by 20% due to the COVID-19 pandemic. An increase in the poverty rate in Kenya by 13% during the pandemic was also reported by Termeer et al. [25]. Similar observations were confirmed later by the household surveys in Kenya, Tanzania, and Namibia [26], as well as the global assessment presented by Béné et al. [27]. Most recently, Balistreri et al. [13] analysed the impact of the COVID-19 pandemic and associated policy responses on the global economy and food security in 80 low- and middle-income countries, using two global economy-wide models. They concluded that the pandemic's main effects were exacerbating the declining food security trend in all the 80 countries covered in their study. Most of the increase in the number of food insecure people from COVID-19 in 2020 was driven by large Asian countries, particularly India, Bangladesh, and Pakistan. Also, Saboori et al. [12] came to similar conclusions. They examined the effect of the COVID-19 pandemic on food security, focusing on factors of affordability, availability, quality and safety, natural resources, and resilience from 2012 to 2020 for 102 countries. The results confirmed that a growing inflation rate had a negative effect on total food security and affordability indices due to the fact that the growth of food prices was much higher than in the case of other commodities. This decreased purchasing power of consumers exacerbated their vulnerability to price growth and reduced food affordability. The study shows that countries with lower food security were more vulnerable to economic shocks, such as the pandemic and inflation than countries with higher food security levels. The paper of Cho [28] confirms the latter observation also in Africa and Southern Asia.

Studies on food security aspects related to the COVID-19 outbreak in developed countries are scarce. An exacerbation of food insecurity in the US during the pandemic was proved by Niles et al. [29] and Parekh et al. [30], while Lauren et al. [31] showed that there are strong associations between being at risk of food insecurity and anxiety/depression. As far as the EU countries are concerned, the food security problem was analysed by Penne and Goedemé [32] in terms of decreased incomes of consumers and the availability of a meal with meat, chicken, fish, or a vegetarian equivalent. The authors concluded that in 16 out of 24 EU countries at

least 10% of the population in (sub)urban areas has been at risk of being confronted with income-related food insecurity. Similar observations were made by Cattivelli [33] in the paper on food insecurity in Italy, which increased sharply during the pandemic, particularly among those who contracted the virus and had to remain in quarantine and those who suddenly lost their jobs. As results from a study on socioeconomic risks of food insecurity in the UK by Brown et al. [34], policies providing additional financial support might help to reduce the impact of COVID-19 on food insecurity. A study by Marchetti and Secondi [35] estimates that in Italy, the problem of being at-risk-of-food-poverty or food insecure concerns up to 22.3% of the entire population, and it increased during the pandemic. Due to the fact that in Europe during the pandemic, the number of people at risk of malnutrition doubled [14], in our paper, we have attempted to identify the effects of COVID-19 in several aspects characterising food insecurity in all the EU countries while filling an important research gap in this area.

## Data and methodology

We used various data to estimate the effect of COVID-19 on changes in the analysed factors affecting food security in the EU. Firstly, the impact of the COVID-19 pandemic on physical food access was investigated by examining food trade dynamics. Secondly, the impact of the COVID-19 pandemic on economic food access was studied by examining monthly and annual changes in food prices, the share of food in inflation, the financial situation of citizens, the share of food expenditure in income as well as the ability to afford a meal with meat, fish or vegetarian equivalent every second day. The abovementioned factors influenced the increased share and number of people who are severely food insecure.

Annual and monthly data on the food trade were derived from the Comext database [36] within the Eurostat resources. Food trade values cover exports and imports of food commodities included in the 01–21 chapters of the Harmonized System Nomenclature. The price analysis was conducted based on yearly and monthly harmonised indexes of consumer food prices (2015 = 100). The time range of the monthly observations used was from January 2010 to February 2022, while the annual ones covered 2000–2021. In addition, an analysis of food weights in the basket of goods covered by the harmonised index of consumer prices (HICP) was conducted. The income situation of the EU citizens was assessed based on the Eurostat [37] annual data on the median equivalised net income expressed in national currencies. The data cover the period of 2012–2021.

To illustrate the state of food security and nutrition, we used data collected in the EU Survey of Income and Living Conditions by Member States (EU-SILC), published by Eurostat [37]. One component of the survey is an indicator that shows the share of people unable to afford a meal with meat, chicken, fish, or a vegetarian equivalent every second day. Previous studies have used this measure as a proxy for household food insecurity [38]. The data covered 2003–2021, although information for 2003–2004 was missing for many countries. Hence, only the period 2005–2021 was analysed. The other measure we examined in the study was the prevalence of severe food insecurity in the total population (in %, 3-year average) calculated by FAO [14].

All analyses (on trade, prices, income, wages, ability to afford a ability to afford a meal with meat, chicken, fish, or a vegetarian equivalent, and prevalence of severe food insecurity) are based on data sets obtained from publicly available databases of the European Union Statistical Office [37] and the Food and Agriculture Organization of the United Nations [14]. The paper's authors did not use the primary, individual data collected with human participants. The summary of data specification is presented in Table 1. It is worth emphasizing that for analytical purposes (including ex-post forecasts), data from the periods included there were used, while

**Table 1. Data specification.**

| Data | Frequency | Period | Unit | Source |
|---|---|---|---|---|
| Export/import data | yearly | 2000–2021 | Mln euro | Eurostat |
| Export/import data | monthly | Jan. 2010-Dec.2021 | Mln euro | Eurostat |
| Index of food prices | yearly | 2000–2021 | 2015 = 100 | Eurostat |
| Index of food prices | monthly | Jan. 2010-Dec. 2021 | 2015 = 100 | Eurostat |
| The share of food in inflation | yearly | 2001–2022 | % | Eurostat |
| Median equivalized net income | yearly | 2012–2021 | Nat. currency | Eurostat |
| The share of food expenditure in income | yearly | 2012–2021 | % | Eurostat |
| GDP per capita in PPS | yearly | 2012–2021 | Nat. currency | Eurostat |
| Average full-time adjusted salary per employee | yearly | 2012–2021 | Nat. currency | Eurostat |
| The share of people unable to afford a meal with meat, chicken, fish, or a vegetarian equivalent | yearly | 2005–2021 | % | EU-SILC/ Eurostat |
| Prevalence of severe food insecurity in the total population | yearly | 2018–2021 | %, 3-year average | FAO |

Source: Authors' own research.

the tables and charts in the Results and Discussion section present only research results for the pandemic period.

To analyse the indicators related to both physical and economic food access, we compared:

1. The level of data (on trade, prices, income, wages, GDP in PPS, ability to afford a meal with meat, chicken, fish, or a vegetarian equivalent, and prevalence of severe food insecurity) in the pandemic years, but prior to the Russia-Ukraine war, which could have made the COVID-related impact less clear (2020–2021), with the pre-pandemic level (mostly as of 2019), and

2. The level of data (on trade, prices, income, ability to afford a meal with meat, chicken, fish, or a vegetarian equivalent) in the years covered by the pandemic (2020–2021) with forecasts of these data obtained from forecasting models estimated based on data recorded before the outbreak of the pandemic (until 2019 or until February 2020).

In the second approach, we assume that COVID-19 could have led to significant changes in indicators reflecting the level of food security in EU countries. The direction and strength of the impact of the pandemic can be concluded by comparing two paths of development of the phenomenon: "COVID-19" and "no-COVID-19". Comparison of the observed time paths of the actual levels of variables (COVID-19) with those that would have been expected in the no-COVID-19 scenario defined by the forecasting model allows estimation of the direct impacts of the pandemic. Differences between scenarios are expressed in the form of ex-post forecast errors in a country $i$ and in time $t$:

$$ERROR_{i,t} = ("COVID-19"_{i,t}/"no-COVID-19"_{i,t})/("COVID-19"_{i,t})*100.$$

Positive error values indicate that COVID-19 contributed to the growth of a given phenomenon, whereas negative values indicate that COVID-19 led to a decline in it. The calculated forecast errors for individual countries and over time were the starting point for further analysis.

The second approach allows for the sole impact of COVID-19 by considering pre-existing time series patterns (trends or seasonal fluctuations). Of course, it also has some disadvantages

since it assumes that the only factor affecting forecast errors in our case is COVID-19. Comparing actual data with counterfactual values derived from forecasting models can provide an alternative to approaches based on simulations or econometric models (e.g., the difference-in-difference model).

Most variables used were annual frequency and relatively short time series (from 2000). For this reason, Holt's exponential smoothing model was used to forecast the hypothetical path of the phenomenon, which allows the trend to be taken into account without a priori assuming its functional form (stochastic trend). Holt's forecast (F) in a period t for the h period ahead is calculated as a sum of two components–the level and the trend change of the time series. The level ($L_t$) and the trend change ($T_t$), as well as forecasts for the following h periods, can be written using the following formulas [39]:

$$\begin{cases} L_t = \alpha Y_t + (1 - \alpha)(L_{t-1} + T_{t-1}) \\ T_t = \beta(L_t - L_{t-1}) + (1 - \beta)T_{t-1} \\ \qquad F_{t+h} = L_t + hT_t \end{cases}$$

where: $\alpha$ and $\beta$ are smoothing constants for level and trend change, respectively. Smoothing constants are limited to the range from 0 to 1 and can be specified through an optimisation process, minimising errors [40].

Moreover, the Holt-Winters model (multiplicative case) for monthly trade values and food price series was used, allowing seasonal variations to be included. It is based on three smoothing equations [39]:

where: $S_t$–a seasonal component at time $t$, $\gamma$ –a seasonal smoothing constant, $r$–a number of seasons in a year, the other notations as in Holt's model. As with Holt's model, smoothing constants are limited to the range from 0 to 1.

## Results and discussion

### Impact of the COVID-19 crisis on trade dynamics

Since March 2020, the world economy has suffered a recession resulting from the rapid spread of the COVID-19 pandemic. This was reflected in the simultaneous increase in policy strictness measured by the Oxford Stringency Index and a decrease in the workplace and retail mobility rates expressed by the Google Mobility indices, which were particularly visible between March and June 2020 [11]. Although a decline in the 2020 real-world export value was expected, ranging from 8.1% (the optimistic scenario) to 20.4% (the pessimistic scenario) [41], that scenario did not come true, and a drop in total trade was 5.3% in 2020 [42]. Strong monetary and fiscal policies in many countries, numerous attempts to stabilise households and business economic activities, as well as trade policy restraints were the most important measures that enabled this [43].

The impact of the COVID-19 pandemic on the EU food and non-food trade is shown in Table 2. Overall, the pandemic had a negative impact on foreign trade in goods from April 2020 to February 2021 which is reflected in negative differences between "COVID-19" and "no-COVID-19" scenarios. As of March 2021, the actual level of trade is at a similar or even higher level, especially for the non-food trade, compared to the predictions obtained from the Holt-Winters model. The negative impact of the COVID-19 pandemic on the food trade was smaller than on the non-food trade. Non-food imports and exports in May 2020 were about 40–45% lower than forecasts, while the decline in agri-food trade at that time was 14–16%. It can also be noted that in the case of non-food trade, the first phase of the pandemic played a key role, while in the case of food trade the decline in exports and imports was comparable in

**Table 2. Differences between actual and predicted series of international trade in the EU countries during the COVID-19 pandemic (%).**

| Specification | | Jan. | Feb. | Mar. | Apr. | May. | Jun. | Jul. | Aug. | Sep. | Oct. | Nov. | Dec. |
|---|---|---|---|---|---|---|---|---|---|---|---|---|---|
| Food import | 2020 | x | x | 0.88 | -5.55 | -14.23 | -5.19 | -3.77 | -7.97 | -1.79 | -6.62 | -6.64 | -4.84 |
| | 2021 | -12.51 | -5.98 | 3.47 | -0.66 | -4.08 | 3.96 | -0.26 | 1.57 | 5.86 | -0.88 | 8.21 | 12.33 |
| Food export | 2020 | x | x | 0.48 | -7.33 | -15.88 | -6.09 | -3.91 | -8.85 | -3.23 | -6.51 | -7.60 | -2.96 |
| | 2021 | -10.79 | -4.73 | 4.22 | -0.33 | -3.68 | 3.51 | -1.71 | -0.18 | 4.62 | -2.88 | 4.76 | 7.22 |
| Non-food import | 2020 | x | x | -12.75 | -30.88 | -39.29 | -19.04 | -10.64 | -10.94 | -5.71 | -7.09 | -1.88 | 1.81 |
| | 2021 | -7.85 | 0.62 | 11.20 | 7.12 | 5.05 | 12.65 | 8.44 | 11.18 | 16.12 | 14.08 | 23.22 | 27.40 |
| Non-food export | 2020 | x | x | -12.91 | -34.77 | -44.61 | -19.39 | -8.75 | -10.64 | -4.27 | -3.73 | -1.62 | 1.56 |
| | 2021 | -5.79 | -0.70 | 7.39 | 5.20 | 0.35 | 8.85 | 5.85 | 5.43 | 10.41 | 7.50 | 15.00 | 17.78 |

Source: Authors' calculations based on [36].

the first and second phases of the pandemic, which peaked in May 2020 and January 2021. The above-described trends can be referred to as the COVID-19 effect resulting from the trade restrictions implemented and transport disruptions.

In agri-food trade, some restrictive trade policy measures were imposed to limit imports from other countries on the one hand and to ensure food availability and avoid increases in food prices on the domestic market on the other hand [11, 44]. However, trade-facilitating policies, including quota expansions or lowering both import tariffs and non-tariff barriers to trade, were also introduced to counteract the possible deterioration of food availability and access [45]. Despite labour shortages, temporary plant closures, and logistics constraints mainly related to additional border controls and sanitary measures, global agri-food trade during the COVID-19 pandemic proved to be relatively stable [10, 45]. Grains and oilseeds markets were unaffected, while beef and pork markets experienced only a temporary export decline [46]. Low-income elasticity of food demand, especially for staple foods, bulk marine shipments not requiring substantial human interaction while being less vulnerable to transport restrictions, along with the key importance of the agri-food industry in government policies, all contributed to the resilience of agri-food trade [47]. The stronger resilience of agri-food trade patterns than in other industries observed for the EU countries is in line with previous studies for global trade by Arita et al. [47] or Engemann and Jafari [11], as well as food trade in the Commonwealth trade by Vickers et al. [48].

Based on the dynamics indices (Table 2) it can be stated that food exports were slightly more negatively affected by the COVID-19 pandemic than imports. Nevertheless, in 2020 the negative errors indicate that food import values in most EU countries were lower during the pandemic than in its absence, while in 2021 import values increased (Fig 1). Food net importing countries such as Malta, Cyprus, Croatia, Greece, Slovenia and Portugal turned out to be the most sensitive to the COVID-induced import declines. Considering the trade position of those countries, a reduction in imports might have threatened food access. This could have been caused by a reduced quantity of imported goods and inflation. On the one hand, a decrease in the food supply in the first months of the COVID-19 pandemic resulted in price increases in the EU, which, along with the deceleration of wage growth, contributed to the deterioration of the economic access to food and rising inability to afford a meal with meat, chicken, fish (or vegetarian equivalent) every second day (Fig 5). On the other hand, food supply disruptions affected product diversity on the market by limiting the availability of fresh and perishable food. Similar short-term effects of COVID-19 on food security and malnutrition in selected Commonwealth countries were indicated by Vickers et al. [48]. All those impacts jointly made the prevalence of severe food insecurity in the total population of most EU countries greater than before the pandemic (Fig 6).

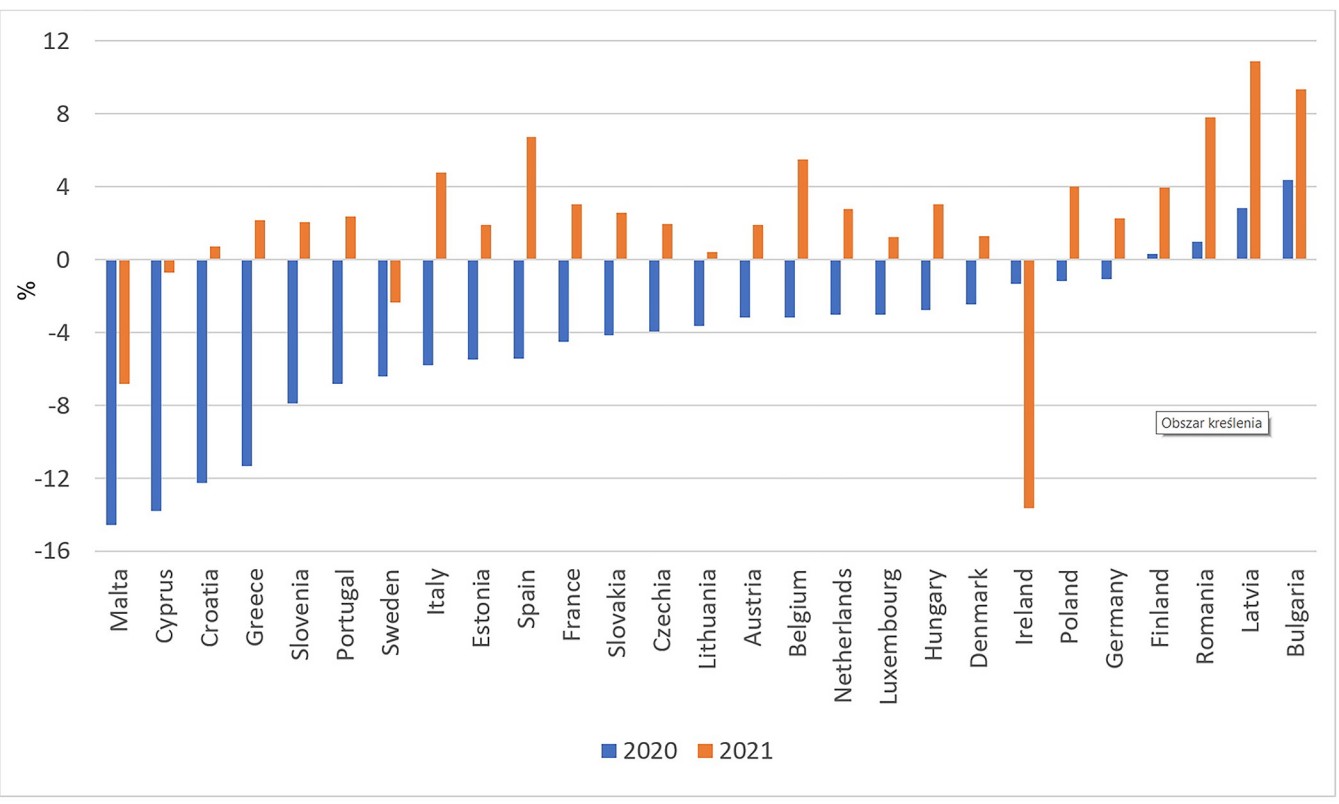

**Fig 1. Differences between actual and predicted series of food imports in the EU countries (%).** Source: Authors' calculations based on [36].

### Impact of the COVID-19 pandemic on food prices

One of the critical factors determining the ability to meet basic human needs is the level of food prices. In light of the Eurostat data, food prices in the EU-27 in 2020 were 2.7% and in 2021–4.2% higher than in 2019. Nevertheless, this is mainly due to general inflation trends. Fig 2 shows forecast errors reflecting the percentage differences between actual prices ("COVID-19" scenario) and predicted prices ("no-COVID-19" scenario according Holt's model) for 2020–2021. In their light, it can be seen that the COVID-19 pandemic contributed to an increase in the prices of food purchased by consumers in the EU-27. Nevertheless, this growth was not strong and amounted to 0.91% in 2020 and 0.67% in 2021. On the one hand, we have countries such as Czechia, Hungary, Bulgaria, and Greece, where food prices were higher than forecasted ones by more than 2%, while on the other hand, there are Estonia, Denmark, and the Netherlands, where prices were below their forecasts.

The annual time series, however, do not fully reflect price dynamics during the COVID-19 pandemic in the EU-27. Therefore, we also calculated percentage errors showing the differences between monthly food price indices and their forecast values based on the Holt-Winters model. In the first step, forecasts were made using data up to February 2020 (before the outbreak of the pandemic), while in the second step, predictions were calculated using data up to February 2022 (before the outbreak of the war in Ukraine). It turned out that the impact of the war in Ukraine on food prices in the EU-27 is much more substantial than that of the pandemic outbreak [49]. Hence, in the study, we do not deal with 2022, as the dominant factor in changes in trade, prices, or food security in that year became the Russia-Ukraine war. In 2020, immediately following the pandemic outbreak, there was an increase in food prices resulting

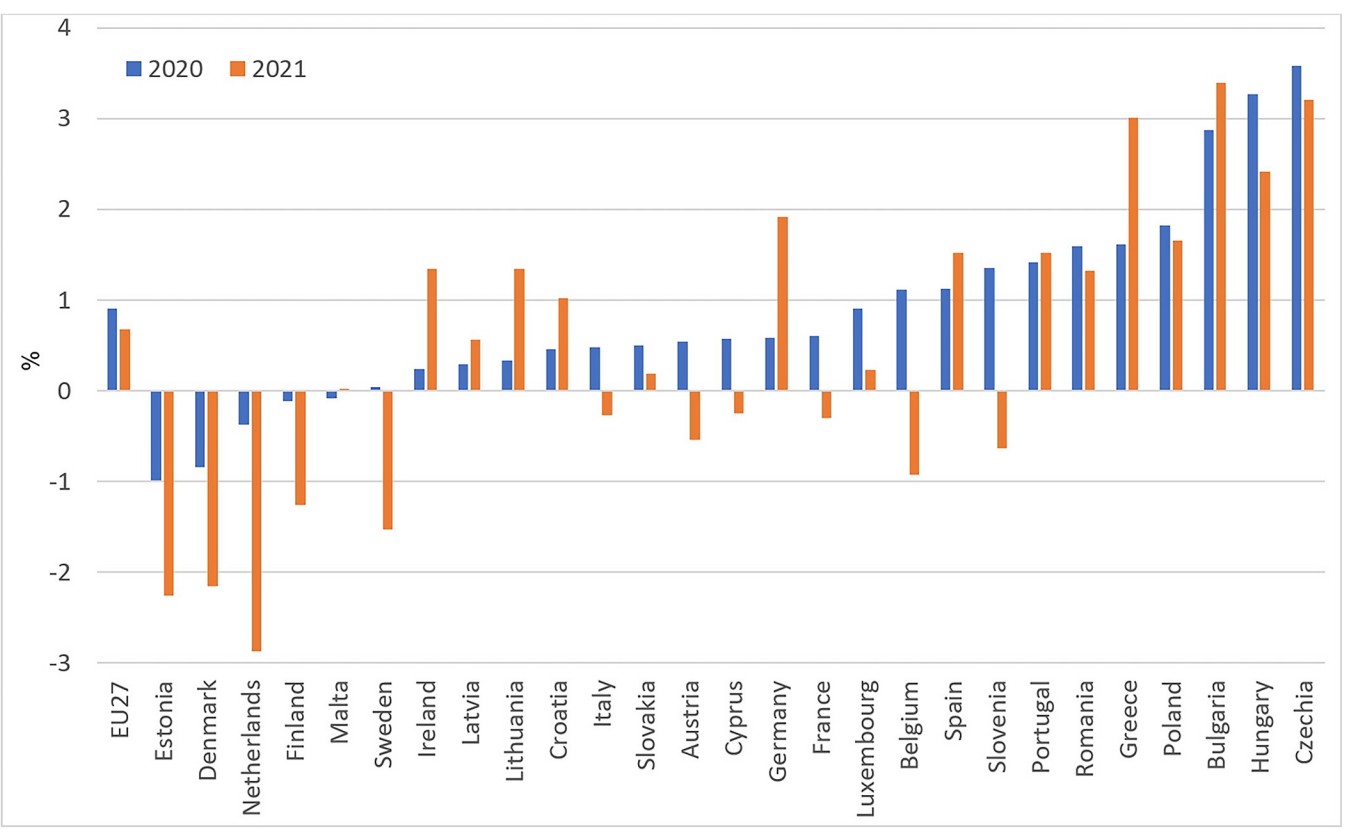

**Fig 2. Differences between actual and predicted series of retail food prices in the EU countries (%).** Source: Authors' calculations based on [37].

from panic food purchasing, as well as accumulation or disruption of supply chains [5]. However, as early as July 2020, prices returned to their long-term path and began to form below it. Our calculations based on Eurostat data [37] show that the rebound in retail prices (the third phase) from March 2021 was sustained and led to price increases of about 4% above forecasts. As explained in the literature, this increase was mainly due to opening borders, reduced inventories, or increased demand due to active state policies [10].

## Impact of the COVID-19 pandemic on consumer food spending, and incomes

Relatively more substantial growths in retail food prices led to increased consumer spending on food in household budgets. This can be seen in changes in the share of food prices in the basket of goods purchased by the consumer during the 2020–2022 pandemic period (Fig 3). At the same time, it is essential to recall that the weights in such a basket for a given calendar year are determined based on consumer spending in the previous year. For example, the weights in 2021 are given based on observations concerning spending in 2020. Thus, the shares in 2021 reflect the level of spending in 2020, while the shares adopted for 2022 are based on consumer spending in 2021. From Fig 3, it can be noted that the pandemic period in the EU-27 saw an increase in the share of food in the basket of goods purchased by consumers relative to the pre-pandemic period by 1.83 and 1.28 percentage points, respectively, in 2020 and 2021. These shares increased the most in Ireland, Spain, Latvia, Italy, Estonia, Portugal, Greece, and Slovakia. It is worth noting that most countries saw declines in these shares over the years preceding

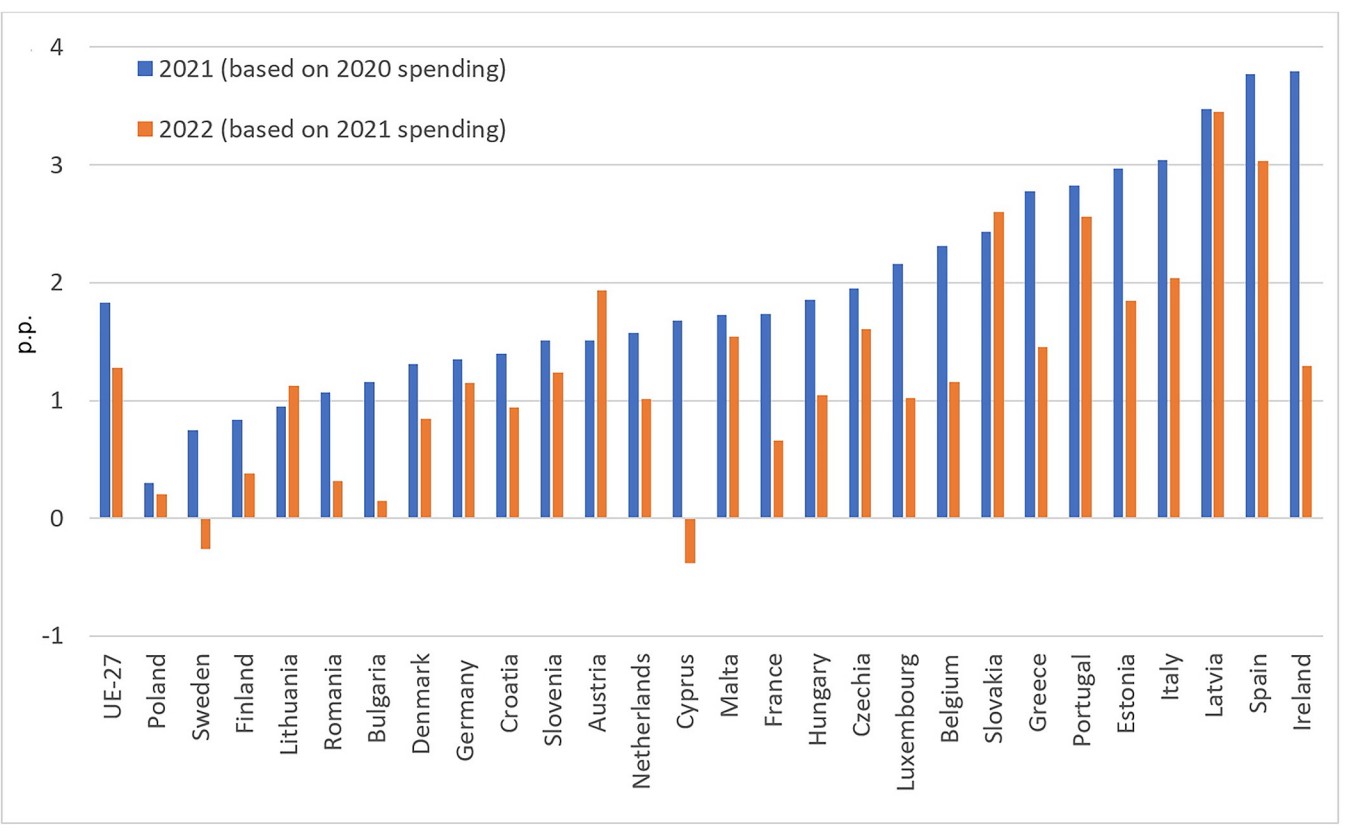

**Fig 3. Changes in the share of food in the inflation basket in EU countries relative to 2020 (p.p.).** Source: Authors' calculations based on [37].

the pandemic. Having in mind that changes in the share of food in total consumer spending at a specific point in time could have been shaped by several factors, it may be stated that, among others, the trend observed over the pandemic years could be related to the increase in disposable income. The latter was possible thanks to social transfers introduced to mitigate the adverse effects of income loss during the pandemic. As shown in Fig 4, the increase in disposable income was greater than the increase in food prices.

An important factor affecting the economic access to food is consumer income, which may offset price increases. Consumer income generally depends on wages and other transfers linked with the general economic situation. In 2020, Gross Domestic Product (GDP) per capita expressed in PPS (Purchasing Power Standards) declined in most EU countries–with the most significant declines in Spain (12.5%), Greece, and Malta (9.4%) [37]. In the EU-27 it fell by 4.1% in 2020. Only five countries recorded slight increases in this indicator compared to 2019. Looking at the development of the EU's GDP components from the income side, taxes on production and imports levied by the government, less subsidies granted, (thus government income) decreased the most–by 16.4% on average in the EU [37]. Also, in the case of average wages, declines were recorded for eight countries (especially for the highly developed countries of Western Europe [37]). But in the case of 22 countries the level of wages in "COVID-19" scenario was lower than in "no-COVID-19" scenario (based on the authors' compilation using Eurostat [37]). These observations indicate the negative effect of the COVID-19 pandemic on wages.

Nevertheless, Eurostat data [37] shows that aggregated disposable incomes, which refer to households' incomes from market sources and cash benefits after deducting direct taxes and

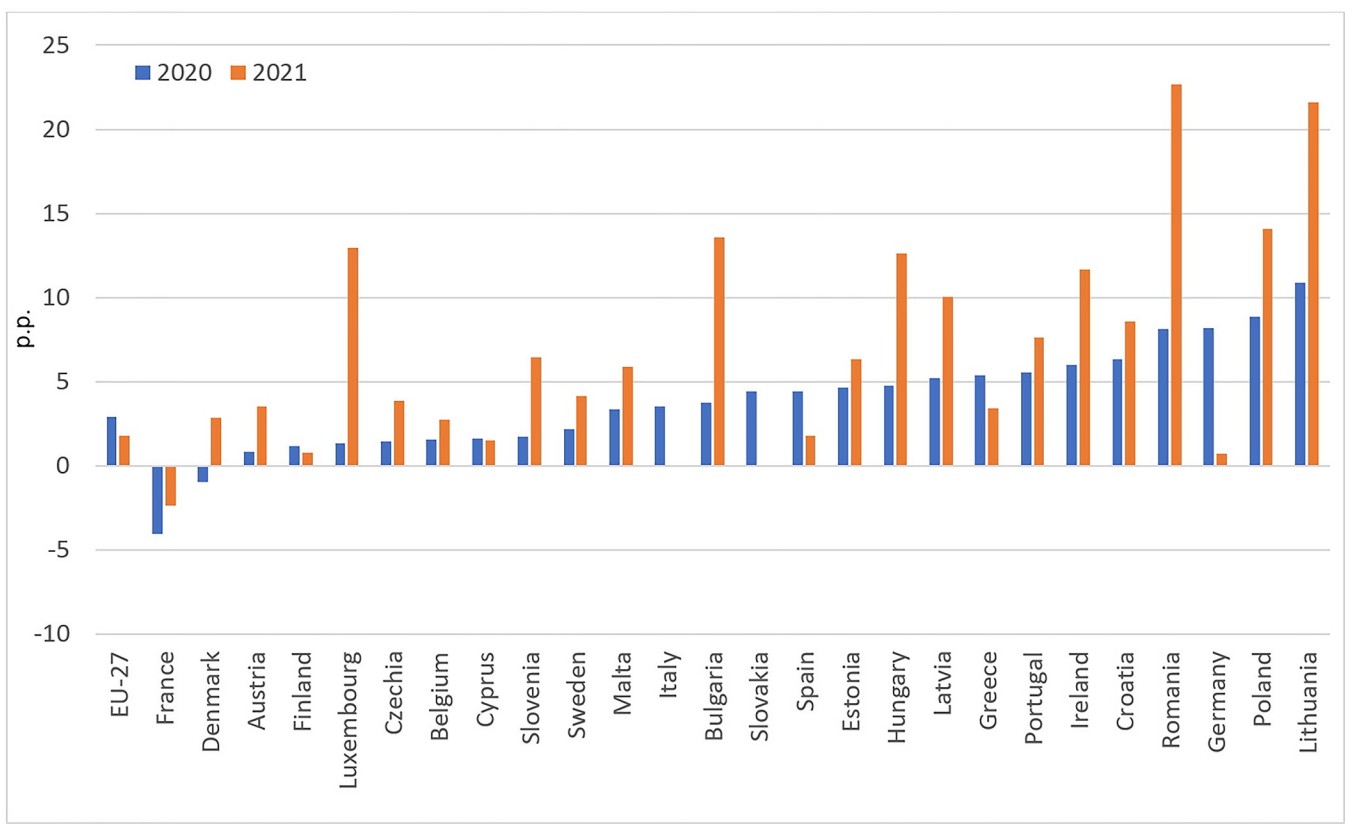

**Fig 4. Differences between median equivalized net income changes and food price changes (measured in national currencies, EU-27 in euro) relative to 2019 (p.p.).** Source: Authors' calculations based on [37].

regular inter-household cash transfers, did not deteriorate. Median equivalized net disposable income in the EU-27 in 2020 was 5.6% higher than in 2019, while in 2021 it was 6.0% higher. It may indicate that household income was sustained to some extent by additional stimulus and safety-net policies, which successfully shielded most people from the recession induced by the COVID-19 pandemic. After the COVID-induced initial job losses in service industries, the EU households were supported by social welfare policies. Almeida et al. [50] also indicate that flexible fiscal policy measures played a significant role in EU countries, reducing the magnitude of income loss and mitigating the impact of the pandemic on poverty.

The question remains: how did disposable income changes relate to the rising food prices. Fig 4 presents the difference between the net income changes and food price changes. In most countries, the increase in disposable income expressed in national currencies was significantly higher than in food prices (Fig 4). France was the only country where food prices rose more markedly than incomes in 2020–2021. Therefore, it can be concluded that the increase in food prices was offset by the increase in disposable incomes, boosted by the social policy transfers.

## Impact of the COVID-19 pandemic on consumer access to the meal with proteins and prevalence of severe food insecurity

Despite an increase in disposable income that outpaced the rise in prices, a sense of insecurity and unfavourable changes in imports had an impact on food deprivation in the EU. In 2000–2019 the share of people unable to afford a meal with meat, chicken, fish, or a vegetarian equivalent every second day steadily declined, except for 2010–2013, when it increased by an average

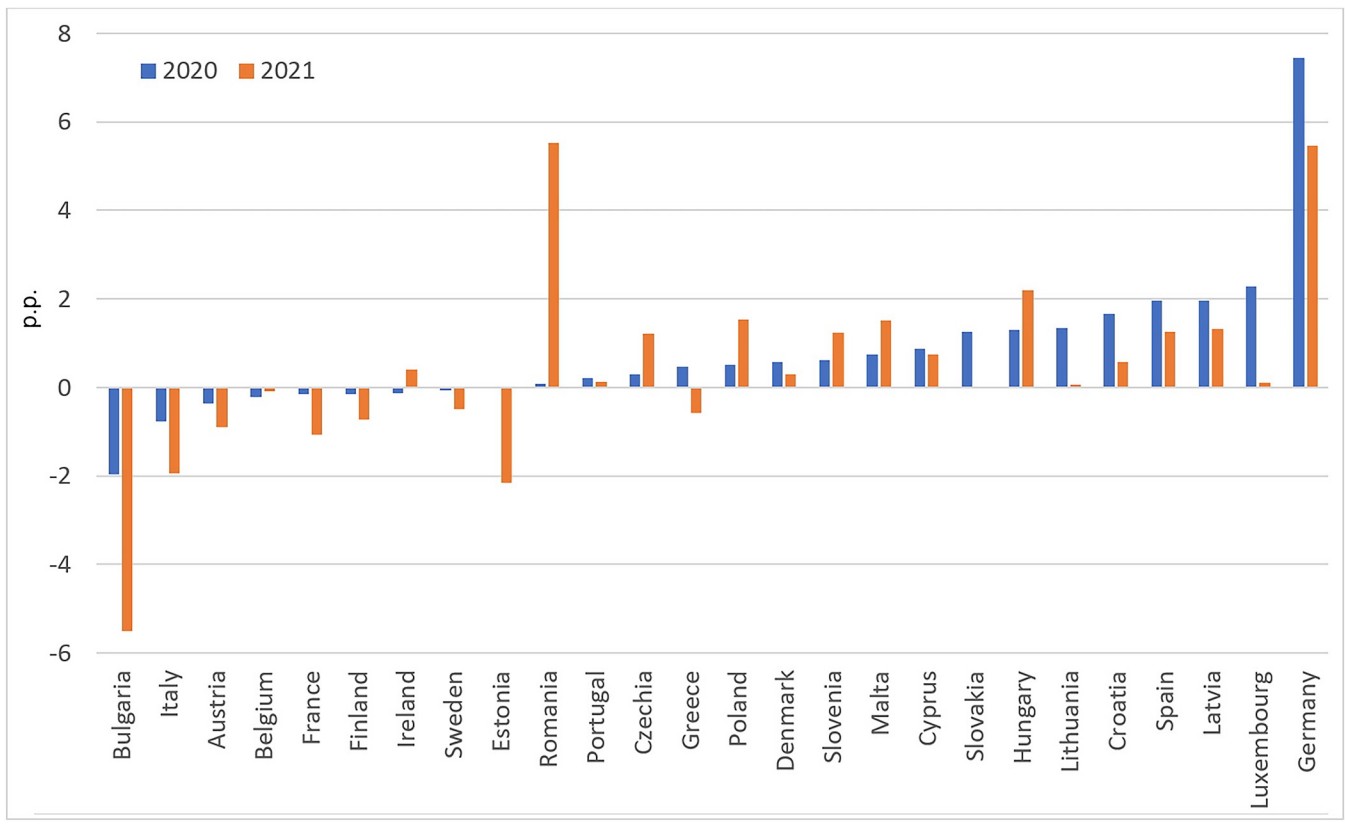

**Fig 5. Differences between the actual and predicted indicator: Inability to afford a meal with meat, chicken, fish (or vegetarian equivalent) every second day (p.p.).** Source: Authors' calculations based on [37].

of 2.7 percentage points. Reeves et al. [38] point out that this may be related to rising cost of food and stagnant wages. In 2020, the indicator increased once again in 13 of the 27 EU countries, while it remained unchanged in 2 states. The share of people unable to afford a meal with meat, chicken, fish, or a vegetarian equivalent every second day in EU-27 aggregate climbed in 2020 to 8.1% compared to 6.8% in 2019. In 2021 it fell to 7.3%; however, the proportion of households reporting the inability to afford a meal with proteins was still higher than in 2019. Moreover, forecast errors (Holt's model; Fig 5) reflecting differences between "COVID-19" and "no-COVID-19" scenarios were positive for 17 countries in 2020 and 16 in 2021 (for 2021, data for Slovakia was unavailable). Surprisingly, the most significant deterioration of the situation was recorded in Germany, one of the wealthiest EU countries.

The increase in food insecurity over the COVID-19 period is also indicated by the FAO index, which provides an estimate for the proportion of the population with severe difficulties in accessing food. As can be seen in Fig 6, there was an increase in the percentage of the population at risk of severe food insecurity in 18 EU countries in 2019–2021 (a 3-year average covering two years of the COVID-19 pandemic) compared to the 2018–2020 period. Notably, increases in the proportion of people experiencing food insecurity only at the severe level and at the moderate or severe level during the pandemic were recorded in all regions of the world [15].

Thus it can be concluded that an observed decrease in the food supply in the first months of the COVID-19 pandemic, limiting the availability of fresh and perishable food and stimulating price increases, contributed to the deterioration of the subjective evaluation of economic access to food, expressed by rising inability to afford a meal with meat, chicken, fish (or

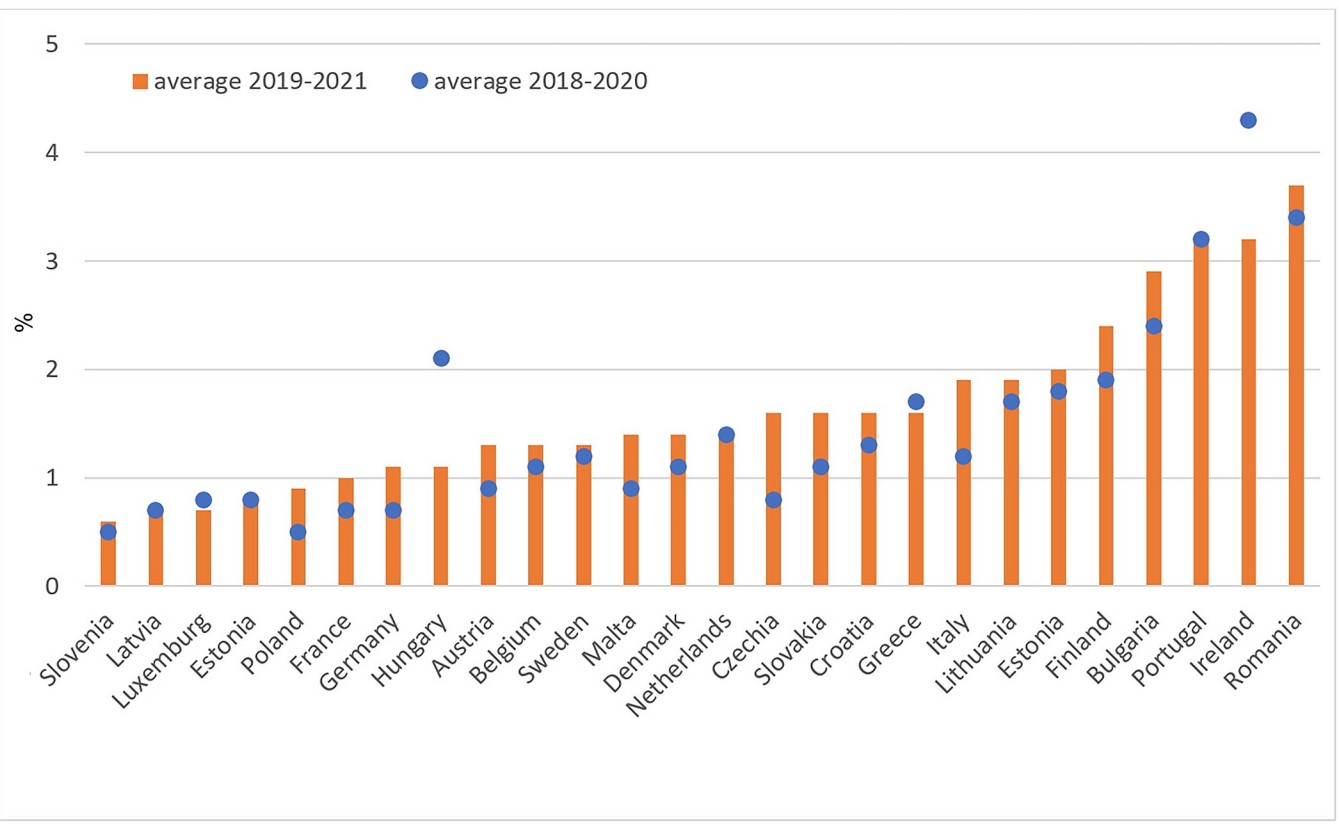

**Fig 6. Prevalence of severe food insecurity in the total population (%, 3-year average).** Source: Authors' calculations based on [14].

vegetarian equivalent). All those impacts jointly, despite of the social transfers which on average boosted the disposable incomes, made the prevalence of severe food insecurity in the total population of most EU countries greater than before the pandemic (Fig 6). The reason for this phenomenon (increase in food insecurity, despite the rising disposable incomes) may be the uneven distribution of poverty and food insecurity in the society, as well as polarised distribution of incomes, which was out of scope of our analysis.

## Conclusions

The aim of the paper was to present an ex-post assessment of the impact of the COVID-19 pandemic on the physical and economic access to food in the EU-27 countries, which translates into the overall state of food security. Trade and price effects of the COVID-19 pandemic, along with the food insecurity and malnutrition indicators, were under investigation. Our research indicates that the EU food trade was more resilient to COVID-19 impacts than the trade in non-food products. Up to 14–16% decreases in food export and import values resulting mainly from the trade restrictions implemented and transport disruptions were of temporary nature and took place only from April 2020 to February 2021. This did not affect the trade balance significantly; however, the reduction in imports threatened the physical food access in most EU countries. The results obtained indicate that food net importing countries were the most sensitive to the COVID-induced import declines.

A comparative analysis of the real data on prices and households' income, as well as the consumer perception of the financial situation and food consumption affordability, does not

offer a clear answer concerning the impact of the COVID-19 pandemic on the EU households food security. Regarding the economic food access, the results show that retail food prices in the EU increased more markedly during the pandemic than previous trends suggest. However, it should be noted here that disposable household income, including social support, rose more than food prices, fully compensating for food price inflation. It may suggest that the COVID-19 pandemic did not significantly affect the deterioration of the economic access to food in the EU countries.

On the other hand, an increase in the share of food in the inflation basket in virtually all the EU countries can suggest the deterioration of households' food security. This was also proved by the analysis of indicators showing the EU citizens' financial situation. A significant increase was observed in the share of the population reporting inability to afford a meal with the recommended amount of animal protein or its vegetarian equivalent every second day. At the same time, according to the FAO data the prevalence of food insecurity increased in most EU countries. It should also be stressed that the conclusions drawn above refer to the average state of food security in all EU households, while the results differed significantly between specific countries.

When interpreting our results, certain limitations should be kept in mind. First of all, it should be stressed that all the variables under investigation were affected by many shocks and the surrounding policies related or unrelated to the COVID-19 pandemic. However, it seems that the COVID-19 pandemic was a major factor affecting households' food security in the years analysed. Moreover, our research was based on the aggregated EU and country-level data, which do not fully reflect the state of food security in households by income status. The abovementioned research limitations could be considered as the basis for further studies.

## Author Contributions

**Conceptualization:** Karolina Pawlak, Agata Malak-Rawlikowska, Mariusz Hamulczuk, Marta Skrzypczyk.

**Data curation:** Karolina Pawlak, Mariusz Hamulczuk, Marta Skrzypczyk.

**Formal analysis:** Karolina Pawlak, Agata Malak-Rawlikowska, Mariusz Hamulczuk, Marta Skrzypczyk.

**Funding acquisition:** Karolina Pawlak, Agata Malak-Rawlikowska, Mariusz Hamulczuk.

**Investigation:** Karolina Pawlak, Agata Malak-Rawlikowska, Mariusz Hamulczuk, Marta Skrzypczyk.

**Methodology:** Karolina Pawlak, Agata Malak-Rawlikowska, Mariusz Hamulczuk, Marta Skrzypczyk.

**Project administration:** Agata Malak-Rawlikowska.

**Resources:** Karolina Pawlak, Agata Malak-Rawlikowska, Mariusz Hamulczuk, Marta Skrzypczyk.

**Software:** Karolina Pawlak, Mariusz Hamulczuk, Marta Skrzypczyk.

**Supervision:** Agata Malak-Rawlikowska.

**Validation:** Karolina Pawlak, Mariusz Hamulczuk, Marta Skrzypczyk.

**Visualization:** Karolina Pawlak, Mariusz Hamulczuk, Marta Skrzypczyk.

**Writing – original draft:** Karolina Pawlak, Agata Malak-Rawlikowska, Mariusz Hamulczuk, Marta Skrzypczyk.

**Writing – review & editing:** Karolina Pawlak, Agata Malak-Rawlikowska, Mariusz Hamulczuk, Marta Skrzypczyk.

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
