## [Decision Letter · Decision Letter 0]

28 Nov 2023

PONE-D-23-22931Has Food Security in the EU Countries Worsened During the COVID-19 Pandemic? Analysis of Physical and Economic Access to FoodPLOS ONE

Dear Dr. Malak-Rawlikowska,

Thank you for submitting your manuscript to PLOS ONE. After careful consideration, we feel that it has merit but does not fully meet PLOS ONE’s publication criteria as it currently stands. Therefore, we invite you to submit a revised version of the manuscript that addresses the points raised during the review process.

I agree with reviewers that the paper examines an interesting research question; however; It is recommended to: (i) further develop intuitive explanations of the main results of the paper; (ii) identify and emphasize the contribution to the existing literature; (iii) improve the communication styles; (iv) develop a theoretical discussion; and (v) reexamine the main hypothesis. Please submit your revised manuscript by Jan 12 2024 11:59PM. If you will need more time than this to complete your revisions, please reply to this message or contact the journal office at plosone@plos.org. Please include the following items when submitting your revised manuscript:A rebuttal letter that responds to each point raised by the academic editor and reviewer(s). You should upload this letter as a separate file labeled 'Response to Reviewers'.A marked-up copy of your manuscript that highlights changes made to the original version. You should upload this as a separate file labeled 'Revised Manuscript with Track Changes'.An unmarked version of your revised paper without tracked changes. You should upload this as a separate file labeled 'Manuscript'.

We look forward to receiving your revised manuscript.

Kind regards,

Jubril Animashaun

Academic Editor

PLOS ONE

“This research has received funding from the National Science Centre within the OPUS research project no. 2021/41/B/HS4/03161 entitled “The implications of the COVID-19 crisis for the spatial integration of agri-food markets and the functioning of food supply chains in the world, with a particular focus on Poland.””

Reviewers' comments:

Reviewer's Responses to Questions

**Comments to the Author**

1. Is the manuscript technically sound, and do the data support the conclusions?

Reviewer #1: Yes

Reviewer #2: Yes

2. Has the statistical analysis been performed appropriately and rigorously? 

Reviewer #1: Yes

Reviewer #2: No

3. Have the authors made all data underlying the findings in their manuscript fully available?

Reviewer #1: Yes

Reviewer #2: Yes

4. Is the manuscript presented in an intelligible fashion and written in standard English?

Reviewer #1: Yes

Reviewer #2: Yes

5. Review Comments to the Author

Reviewer #1: Why is the Covid-19 pandemic viewed as a climate-environmental shock by the author as one must conclude from the text in ll 59-64? It has been a public health phenomenon, not related to climate. It affected the environment, but not in the context of the climate change as the text would suggest.

l. 94; do you mean “food access”? Not sure what is meant by “economic food access” (what is the difference between food access and economic food access?). Is it “affordability”? Clarify. See your discussion in ll 180-188.

l. 104; do you have the data to make inferences about the “healthy diet”? Or, is the focus only on household ability to purchase food (affordability) and disruption of trade flows?

Section 2; you discuss the actual epidemics (for example, the bird flu), not the threat of epidemics. Use the term “epidemic”.

Ll 141 and following. There was no outbreak of BSE in the USA. There were very few cases and they were handled expeditiously. What did happen, was the ban of US beef in several major importing countries and, given the dependence on beef exports, the cattle and beef industry in the US was affected.

Ll 150-151; advice caution in making such statements. Meat affordability is a problem in low income countries, which are more likely to consume fowl, rather than pork. The observation may be applicable to the middle-income countries. In low income countries, consumers depend on plant protein sources.

l. 159; was it the loss of income or the loss of purchasing power due to inflation? What has caused incomes to decline? Loss of jobs? Clarify.

l. 159 and following; use consistently the same terminology; is Covid-19 the same as SARS…? If yes, choose one term and use in the paper. Similarly, l. 170, Covid-19 pandemic or crisis? Use one term only.

Ll 211-213; do not use one-sentence paragraphs. Merge with the following paragraph.

Ll 213-217; note that now you reversed the order listed in ll 94-101, where the household ability to afford food was the first objective, and the trade flow disruption was the second goal. By consistent in your presentation and analysis. The following text reversed the order again. This is confusing.

Ll 218-222; you seem to mix the analysis of affordability and the trade flows: “…as well as other indicators related to the physical food access (such as trade dynamics)…” Please, clarify.

L. 223 and l. 226; replace the word “variables” by stating clearly what factors you consider. Also, is the word “or” following 2) correct, or should it be “and”. Ll. 229 and following; the discussion gets confusing. It seems that you are now pondering what to do. This is the section about methodology and by now, it must be clear in your approach what you do and present the data and the methods.

The sentence (starts in l. 236): For example, positive error values indicate that COVID-19 contributed to the growth of a given phenomenon during that time, is not clear.

l. 239; “most” variables are annual observations or all of them? Such a statement without presenting the data first confuses the reader; it is not clear what you refer to.

In l. 239, there is a reference to prices, but the factors named earlier included income, wages, etc. beside prices. The presentation must be reorganized to assure clarity.

l. 280; here the references is made to annual and monthly observations. Clearly separate the section into the presentation of the data and then discuss the applied methodology in the context of the available data because it seems the nature of the data dictates the choice of analytical tools, not the other way round.

l. 239 and ll 302-303 present contradiction – are the data since 2000 or since 2005? You need to clearly state what is the length of an observation (monthly or annual for what data) and what data are for what period (2000, after 2005?). A clear presentation of each variable series, with the length of observation, period, and definition of units in a table and supported by a description will greatly improve the coherence of the presentation.

Table 1 and its description; it appears that the model overestimated (the predicted value > the actual value, so the reported figure is negative) the decrease in non-food trade monthly values in 2020 and underestimated the decrease in 2021. The pattern in forecasting the food trade changes was the same as in the non-food trade in 2020, but less consistent in 2021. This is what Table 1 shows and suggests greater volatility in food trade. One would prefer a model that is consistent (either over- or under-estimates) over a model that yields variable predictions.

All figures lack the indication of the Y axis. Please, mark Y-axis on each figure.

l. 364 and following; there is lack of clarity: the title of Table 1 suggests that the figures in the table are the differences between the actual and predicted values. In l. 362, you suggest that the smaller negative values in the case of food trade indicate that the trade was less affected, but is this the case? We do not know the absolute values (is it in money rems or weight?) of the actual trade to make such an inference. The figures in Table 1 show the difference between the actual and predicted values and ignore the volume. If the actual trade declined from month to month, then even if the differences are small, the actual declines could have been large enough to compromise food security. If the values used in the analysis are in money terms, then the inflation may have hid the size of the decline in weight. There is some relevant discussion missing.

Ll 362-378; you may want to clearly delineate between your findings and the references to other studies, especially in the context of diet quality. Your analysis does not provide (at least until this point) any measure of trade composition that would indicate the change in quality of the diet.

Review your use of the term “trend” or “trends” in various places in the paper. Trend is a statistical term with a specific meaning that, in my view, does not fit the context in which you used it.

Please use Russia-Ukraine war, not “conflict in Ukraine”. The latter is misleading and suggests this is an internal Ukraine problem which is not true.

l. 406; re-write.

Ll 407-410; the last sentence, referring to [10], is used as the explanation of your results. However, the connection is tentative because those are two different studies. Results in [10] may coincide with your results, but they do not outright explain them.

L 414 and following; remind the reader what data you use to discuss the basket of consumer goods (from what period?) and the specific method used. Again, caution the reader because the comparison is based on information collected at distant points in time and could have been shaped by several factors. You attribute the observed differences to the Covid-19 pandemic without any hard evidence.

Ll436-437; income is not linked to economic growth. Income changes may be linked to economic growth or decline. Be precise.

L 437 and following; PPS is affected by each country’s rate of inflation.

l. 441; is it -5.1% and -1.1%?

The discussion in ll 443-449 is disconnected from the earlier discussion of the PPP. PPP and disposable income are two different concepts and cannot be compared. From the household perspective, it is PPP that decides the household budget constraint and ability to purchase food. Only in this context can you place comments about the disposable income, which although increased, apparently not enough to offset the decline in PPP. The disposable income may have increased due to government policies such as suspending some taxes or direct transfers, but PPP decreased because of the rising price level. Clarify those issues and their distinction.

L 456; you just showed that PPP has declined, despite larger disposable income. It is not clear what is the purpose of the discussion in ll 456-468. You bring a completely new issue, the social depravation without defining what do you mean. I suggest to delete the whole paragraph.

The section is confusing. In L. 482 you mentioned disposable income, but it is PPP that really measures the ability to buy. You concluded the PPP has declined, so how relevant is the increase in the disposable income? You cited a reference earlier with regard to the fiscal policies – I commented on the issue above – a suspension of selected taxes or direct transfers to individuals could have increased the disposable income and partially offset the price increases (captured in PPP deterioration). Consumption of animal protein is your selected indicator, but it is only one factor. Consumers have been documented to substitute one food type for another. For example, increasing the consumption of yogurt or curd provides animal protein and replaces meat. You notice that the period 2010-2013 was different and that period was associated with the global financial crisis of 2009/2010. You can make very tentative statements and I wonder if they really make a meaningful difference in your paper.

Reviewer #2: The authors made a bold attempt to investigate how COVID-19 affected Food Security in the EU. The research is novel in that only few studies have delved into this regional analysis with most of the literature cited carrying out their research at country-level. It is commendable that the Russian-Ukranian war is factored into the approach that the authors engaged. However, I have strong reservation with the author's attempt to subjectively analyse food security based on consumers' perception. Food security concept has clearly established quantitative and qualitative indicators that can be used and as such should not be based on perception or questions around "make ends meet". In my opinion, this poses a major flaw for the manuscript, hence needing revision. I also suggest that the authors need to rein in their review of literature to make this more centered on the main points of their research and for a better flow. The manuscript may also benefit from grammatical improvements in some areas. For instance, line 105-106. To address the grammatical errors, I suggest authors use a grammar editor tool such as grammarly. If the authors are unable to primarily examine the food security, the manuscript may be better off without the consumers' perception of financial situation and healthy diets section because technically the section fails to address their objective with respect to food security.

6. PLOS authors have the option to publish the peer review history of their article (what does this mean?). If published, this will include your full peer review and any attached files.

Reviewer #1: No

Reviewer #2: No

---

## [Author Response · Author response to Decision Letter 0]

10 Jan 2024

We have attached the "Response to Reviewers" file in the Upload section. All answers to Editor and Reviewers were included there.

---

## [Decision Letter · Decision Letter 1]

27 Mar 2024

Has Food Security in the EU Countries Worsened During the COVID-19 Pandemic? Analysis of Physical and Economic Access to Food

PONE-D-23-22931R1

Dear Dr. Malak-Rawlikowska,

We’re pleased to inform you that your manuscript has been judged scientifically suitable for publication and will be formally accepted for publication once it meets all outstanding technical requirements.

Kind regards,

Muhammad Khalid Bashir, PhD

Academic Editor

PLOS ONE

Additional Editor Comments (optional):

Authors need to change two expressions throughout the paper:

1. "salaries" to "wages"

2. "Percentage points" to "percent".

Reviewers' comments:

Reviewer's Responses to Questions

**Comments to the Author**

1. If the authors have adequately addressed your comments raised in a previous round of review and you feel that this manuscript is now acceptable for publication, you may indicate that here to bypass the “Comments to the Author” section, enter your conflict of interest statement in the “Confidential to Editor” section, and submit your "Accept" recommendation.

Reviewer #1: All comments have been addressed

Reviewer #2: All comments have been addressed

2. Is the manuscript technically sound, and do the data support the conclusions?

Reviewer #1: Yes

Reviewer #2: Yes

3. Has the statistical analysis been performed appropriately and rigorously? 

Reviewer #1: Yes

Reviewer #2: Yes

4. Have the authors made all data underlying the findings in their manuscript fully available?

Reviewer #1: Yes

Reviewer #2: Yes

5. Is the manuscript presented in an intelligible fashion and written in standard English?

Reviewer #1: Yes

Reviewer #2: Yes

6. Review Comments to the Author

Reviewer #1: Change "salaries" to "wages" in the paper. "Percentage points" to "percent". I do not have additional comments.

Reviewer #2: The authors have made substantial effort to address the issues I raised on their initial submission.

7. PLOS authors have the option to publish the peer review history of their article (what does this mean?). If published, this will include your full peer review and any attached files.

Reviewer #1: No

Reviewer #2: **Yes: **Toyin Benedict Ajibade

---

## [Editor Report · Acceptance letter]

5 Apr 2024

PONE-D-23-22931R1 

PLOS ONE

Dear Dr. Malak-Rawlikowska, 

I'm pleased to inform you that your manuscript has been deemed suitable for publication in PLOS ONE. Congratulations! Your manuscript is now being handed over to our production team.

Kind regards, 

on behalf of

Dr. Muhammad Khalid Bashir 

Academic Editor

PLOS ONE